# Full-Scale Fatigue Test and Finite Element Analysis on External Inclined Strut Welded Joints of a Wide-Flanged Composite Box Girder Bridge

**DOI:** 10.3390/ma16103637

**Published:** 2023-05-10

**Authors:** Bin Wang, Laijun Liu, Yuqing Liu, Xudong Jia, Xiaoqing Xu, Kaixiang Miao, Jiandong Ji

**Affiliations:** 1School of Highway, Chang’an University, Xi’an 710064, China; 2017021033@chd.edu.cn (B.W.);; 2Shanxi Jiaoke Highway Survey and Design Institute Co., Ltd., Taiyuan 030032, China; 3College of Civil Engineering, Tongji University, Shanghai 200092, China; 4Shanxi Transportation Survey & Design Institute Co., Ltd., Taiyuan 030032, China; jiaxudong1974@163.com (X.J.); wbinbridge@163.com (J.J.)

**Keywords:** wide-flanged composite box girder bridge, nominal stress method, finite element method, fatigue life, full-scale model test, parametric analysis

## Abstract

For a wide-flanged composite box girder bridge, the risk of fatigue cracking in the external inclined strut welded joint under the fatigue vehicle load is a problem. The main purposes of this research are to verify the safety of the main bridge of the Linyi Yellow River Bridge, a continuous composite box girder bridge, and to propose suggestions for optimization. In this research, a finite element model of one segment of the bridge was established to investigate the influence surface of the external inclined strut, and, using the nominal stress method, it was confirmed that the fatigue cracking of the welded details of the external inclined strut was risky. Subsequently, a full-scale fatigue test of the external inclined strut welded joint was carried out, and the crack propagation law and S-N curve of the welded details were obtained. Finally, a parametric analysis was conducted with the three-dimensional refined finite element models. The results showed that the welded joint in the real bridge has a fatigue life larger than that of the design life, and methods such as increasing the flange thickness of the external inclined strut and the diameter of the welding hole are beneficial to improve its fatigue performance.

## 1. Introduction

Composite box girder bridges have the advantages of being lightweight, easy to fabricate, and high-strength, so they are widely used in the design of large span bridges [1,2,3]. With the increase in traffic volume and vehicle weight, the width of the bridge deck gradually increases, and the length of cantilever arm of the composite box girders also gradually increases. Thus, a wide-flanged composite box girder with long cantilever beam and external inclined struts was developed [4,5]. A wide-flanged composite box girder bridge has a large inclination angle of the external inclined strut and a complex structure of the joint connecting the external diagonal bracing and the crossbeam. Under the vehicle fatigue load, the internal force range of the strut is large, mainly in compressive force cycles. At present, the commonly used joint type is a welded gusset plate connection [6,7,8], which has the advantages of easy fabrication and effective force transmission, but there is a risk of fatigue cracking under repetitive vehicle loading [9,10].

To study the fatigue performance of the external inclined strut, methods such as finite element simulation and the full-scale test were used. Wide-flanged composite box girder bridges have the risk of fatigue damage at the welded details of the gusset plate connection of the external inclined struts. Thus, full-scale fatigue tests on the welded details are important for designing such bridges, which can provide useful information such as the failure mechanism and fatigue life and help to establish guidelines for structural designers [11]. The reduced-scale test has the advantages of being cost-saving, having lower requirements on the test site, etc. However, up to now, finite element simulations or reduced-scale model tests of the external inclined strut were conducted to investigate the fatigue performance of similar welded details in previous research, but full-scale tests were rarely reported [12,13,14], which may also distinguish this research from previous studies.

Currently, gusset plate fatigue problems are becoming increasingly common in steel bridges, and a lot of research has been carried out on the gusset plate fatigue damage problem in steel box girder bridges. Wang [15] carried out the static test on a reduced-scale integral joint of the Wuhu Yangtze River Bridge, analyzed the fatigue life of the fatigue-sensitive details (butt welds, fillet welds, and the plates with chamfers), and used the nominal stress method combined with the S-N curves provided by the American Highway Bridge Design Code AASHTO and the British BS5400 code. Li et al. [16,17] determined the fatigue stress range by a finite element analysis and predicted the fatigue life of the steel beam’s integral joints based on the fracture mechanics equation for a type I crack. Taking the longitudinal inclined strut of the Nanjing Yangtze River Second Bridge as an example, Liu et al. [18] analyzed the fatigue reliability of the gusset plate connection and compared the maintenance methods of drilling stop holes, welding a cover plate, and switching to bolted gusset plate connection through a 3D finite element analysis. Liu et al. found that the method of switching to a bolted gusset plate connection significantly improved the fatigue life. Luo et al. [19] carried out tests on large-scale specimens to obtain the different fatigue life data for the slotted circular hollow sections of the tube-to-gusset plate connections with a coped hole, the specimens were subjected to compressive stress cycles, and the authors numerically simulated the model by the effective notch stress method and recommended measures to improve the fatigue life such as using full-penetration welds instead of fillet welds. Baptista et al. analyzed the gusset plate connection details of circular cross-sections in the existing literature and gave proposals for the classification of the fatigue details. Nassiraei et al. [20,21] assessed the stress concentration factors in tubular T/Y-joints reinforced with a fiber-reinforced polymer using finite element models, carried out parametric studies to investigate the effect of fiber-reinforced polymer, and had a discussion on the S-N curve and the fatigue life of the joints.

In this paper, the design problem of the external inclined strut welded joint in the main bridge of the Linyi Yellow River Bridge is taken as the background. Through finite element simulation and full-scale model testing, the fatigue performance of the gusset plate connection of a wide-flanged composite box girder with an external inclined strut with a square steel tube cross section and the structural reliability of the fatigue details in the real bridge are discussed. Finally, methods to improve the fatigue performance of the welded details are investigated through finite element parametric analysis. The main objective of the simulation and experiment is to verify the safety of the external inclined strut structure of the Linyi Yellow River Bridge and propose optimizations for the structure.

## 2. Background

The main bridge of the Linyi Yellow River Bridge is a (112 + 28 × 128 + 120) m continuous composite box girder bridge. The main girder was constructed by the incremental launching method, divided into two parts each with a 1912 m launching length. The superstructure is a composite structure which is a composite box girder with a transverse frame and precast concrete deck slabs, as shown in Figure 1. The steel box girder is made of weathering steel, with a width of 11 m, a height of 6 m, and transverse cantilevers of 7.5 m on both sides. The precast concrete deck slab is 28 cm thick and 26 m wide. External inclined struts are provided every 4 m on the composite beams.

The external inclined strut is made of a square steel tube with a side length of 320 mm; the thickness of the strut flange connected with the gusset plate is 20 mm; the thickness of the gusset plate is 24 mm; the diameter of the welding hole is 70 mm; and the detail of the external inclined strut section and the welded detail are shown in Figure 2. The steel used is Q345qDNH, and the material properties of the steel used in the bridge construction are shown in Table 1 below. In addition, the steel tube and gusset plate are welded using full-penetration groove welding, with an internal quality that, when examined through ultrasonic flaw detection, should meet the requirement of Grade I in *Non-destructive testing of welds–Ultrasonic testing–Techniques, testing levels, and assessment (GB/T 11345-2013)* [22] and should be in accordance with the *Code for Structural Steel Welding (GB 50661-2011)* [23].

## 3. Determination of the Most Unfavorable Welded Detail of the External Inclined Strut Welded Joint

A finite element model of one segment of the bridge is established, and then the axial force influence surface of the external inclined struts is calculated to determine the position of the most unfavorable external inclined strut. Finally, the stress range of the external inclined struts under the fatigue load is calculated, and the most unfavorable welded details are determined by the nominal stress method.

### 3.1. Finite Element Model of Bridge Segment

A finite element model of the bridge’s one single span was established, as shown in Figure 3. The composite box girder was modeled by S4R shell elements with a global mesh size of 200 mm × 200 mm. The concrete slabs were modeled by C3D8R solid elements with a global mesh size of 100 mm × 100 mm × 100 mm. An S4R element is a four-node linear shell element with one point of numerical integration. A C3D8R element is an eight-node linear solid element with one point of numerical integration. All the elements used are the reduced integration with linear material properties. The constraints of Tie was applied between the surfaces of the concrete and steel elements. The bridge’s segment’s model includes a total of 398,874 elements and 500,058 nodes. The finite element software Abaqus was used for the solving. The static, general step procedure was applied, the incremental number was 1, and the nonlinear effects of geometry were not considered during preprocessing. The calculation showed a good convergence.

The elastic modulus, Poisson’s ratio, and mass density of the steel material were 210 GPa, 0.3, and 7.85 × 10^−9^ t/mm^2^, respectively. The C50 concrete used in the bridge has an elastic modulus, Poisson’s ratio, and mass density of 34.5 GPa, 0.2, and 2.6 × 10^−9^ t/mm^2^, respectively. The fatigue vehicle model III in the *Design Code for Highway Steel Bridges* (*JTG D64-2015*) [24] was used for the fatigue assessment. For calculating the influence surface, a unit load moving over the entire concrete top surface was used. The number of spans of the main bridge in this project was up to 30. However, it is not necessary to model the whole bridge. From the preliminary simulation results of the single-span model, it was observed that the axial force of the inclined strut under the vehicle load is almost the same as when the girder was supported by fixed support or simple support. Therefore, two end supports of the model were fixed, and symmetric restraint was applied to the longitudinal symmetry plane.

### 3.2. Calculation Results

#### 3.2.1. Axial Force Influence Surface

The influence surface of the inclined strut’s axial force was calculated using the finite element model. The values of the axial influence surface of the external inclined struts at 1/4 span, 1/2 span, 3/4 span, and at the support were very close; the influence surface of the struts at 1/4 span, as an example, is shown in Figure 4. In Figure 4, the centerline position of the composite box girder corresponds to *x* = 0 in the transverse direction, and the position of middle-span corresponds to *y* = 0 in the longitudinal direction). The axial force of the external inclined strut sharply reduced as the fatigue vehicle load moved away from the considered external inclined strut. In addition, the calculation results showed that the maximum value of the axial tensile force was two–three orders of magnitude lower than that of the compressive force, so the following calculation of the stress range of the inclined strut ignored the influence of the tensile axial force, and the maximum nominal compressive stress was taken as the stress range in the following full-scale fatigue test.

#### 3.2.2. Determination of the Fatigue-Sensitive Detail and Nominal Stress Range

As shown in Figure 5, four welded details where the structural geometry changed significantly and were prone to stress concentration were sequentially labeled as A, B, C, and D. The most unfavorable loading position of each welded detail was determined by applying four-axis wheel loads on the influence surface. The nominal stress history of the upper and lower welded joints was similar. The curves of the stress history of both detail A and D were bimodal curves. Taking the compressive stress as negative, the nominal stress reached the minimum value when the two axes of the vehicle were loaded symmetrically in the middle span section, and the maximum nominal stress value was nearly 0. Thus, the minimum nominal stress could be used as the compressive stress range. The external inclined strut was under large eccentric compression, and the stress amplitude caused by the axial force only *σ*_n_ was 2.12 MPa. The upper welded joint, which detail A belongs to, was subjected to a larger bending moment. Therefore, the absolute value of the total compressive stress in detail A due to bending and compression was slightly larger than that of the lower welded joint, as shown in Figure 5. Its maximum stress range *σ*_nom_ was calculated to be 6.70 MPa, and the compressive stress due to compression only accounted for 32% of the total stress. In the following section, detail A is studied further.

Further, the Mises stress distribution (in MPa) on the inside and outside flanges of the inclined strut was obtained by the finite element simulation and is shown in Figure 6. The Mises stress *σ* is an effective stress defined by Equation (1). Surrounding the four details, localized stress peaks occurred, and the stress gradient increased significantly. The results indicated that the welded details do change the transmission route in the strut flanges, and the stress concentration was particularly significant in detail A.
(1)$$\sigma =\sqrt{\frac{1}{2}[{\left(\sigma_{1}-\sigma_{2}\right)}^{2}+{\left(\sigma_{2}-\sigma_{3}\right)}^{2}+{\left(\sigma_{3}-\sigma_{1}\right)}^{2}]}$$

## 4. Fatigue Test of the Full-Scale External Inclined Strut

### 4.1. Experimental Program

#### 4.1.1. Specimen Design

To verify the safety of the welded detail of the external inclined strut welded joint under the fatigue vehicle load, a full-scale specimen S1 of a part of the external inclined strut was fabricated according to the geometry of the welded detail in the bridge. The height of specimen S1 was 1364 mm, and the steel material used was the same as that used in the bridge. To facilitate fatigue loading, a top plate, a bottom plate, and two gusset plate flanges were added in addition to the steel tube and gusset plate. The size of the top plate was 552 mm × 400 mm, the size of the bottom plate was 760 mm × 760 mm, the size of the gusset plate flange was 857 mm × 400 mm, the thickness of the top plate and bottom plate was 32 mm, and the thickness of the flange was 16mm. Three views of the specimen are shown in Figure 7.

#### 4.1.2. Loading Scheme

The compressive stress range on the side of the eccentricity in the test was 53.6 MPa, which was about eight times that of detail A in the finite element model of the bridge’s segments. The stress ratio was selected as 0.1 with 4Hz sine wave loading. For the ease of load control, the load range was 720 kN, and the peak load was 800 kN. The loading diagram of the period is shown in Figure 8a. Finally, the calculated ranges of the nominal compressive stress on the side of the eccentricity and the other side were 54.6 MPa and 15.2 MPa, respectively. The specimen was subjected to a total of 2 million cycles of loading with the aforesaid force range as well as three sets of 300,000 cycles for the beachmark test. The magnitude of compressive stress range during beachmark testing was lowered to 80% of the original one, as shown in Figure 8b. In addition, the number of cycles of the beachmark test could be equivalent to 98,000 loading cycles with the original stress range using Miner’s formula [25,26].

The specimen was loaded by an MTS machine, as shown in Figure 9. The bottom of the specimen was bolted to the ground through the bottom beam and anchors, and its upper part was loaded by the distribution beam, which was connected to the actuator. To simulate the loading state of the joints in the bridge, the S1 specimen was eccentrically subjected to a compressive load with an eccentricity of 58 mm, which was consistent with the finite element calculation results of the segmental model.

#### 4.1.3. Setup and Instrumentation

Strain gauge rosettes were placed near the welded details of the specimen to monitor the stress state, as shown in Figure 9. Among them, strain gauges L1 and L2 were used to monitor the stress at the weld toe on the flange of the steel tube, and strain gauges L11 and L12 were used to monitor the stress at the weld root. Simultaneously, a digital magnifying camera was used for crack observation during fatigue loading. The distance measuring function of magnifying camera was used to measure the length and width of the crack.

**Figure 9 materials-16-03637-f009:**
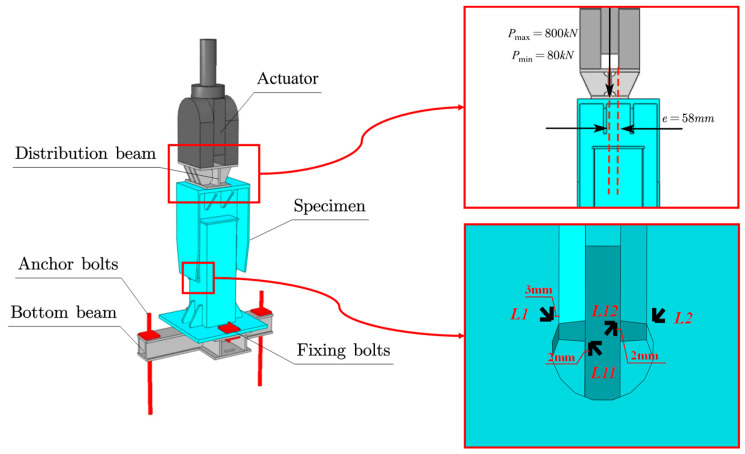
Perspective view of specimen setup.

### 4.2. Experimental Results

#### 4.2.1. Results of Crack Observation

After 700,000 cycles, a fatigue crack was found by the naked eye on the side of the eccentricity, near strain gauge L12. The location where the crack appeared is shown in Figure 10. The crack originated at the mid-face of the gusset plate near the weld root, it developed and extended to both sides in the plate’s thickness direction, and the crack did not penetrate through the gusset plate when loading was completed. The evolution of the crack width and length is shown in Figure 11. This fatigue crack may be related to the initial defects in the base material resulting from the manufacturing process. Fatigue cracks did not appear on the opposite side of the eccentricity.

#### 4.2.2. Stress Results

The relationship between the stress measured by the strain gauges and the number of loading cycles is shown in Figure 12. Figure 12 shows that the principal compressive stress does not vary significantly with the loading cycles, while the principal tensile stress shows a trend of decreasing on L1 and increasing on L2. This may be related to the formation of the fatigue crack. The fatigue crack could transfer compressive stress well and, thus, had less influence on the state of the compressive stress of the welded detail. However, it changed the path of the tensile stress flow, so the principal tensile stress became redistributed.

The stress at the weld root (L11 and L12) was only about 60 MPa, and the stress at the outer weld root (L12) was larger than that at the inner one (L11). Moreover, L12 strain gauge was broken when the number of loading cycles was between 200,000 and 300,000. Since the crack was generated exactly under this strain gauge, the fatigue crack initiation life was determined to be 300,000 cycles.

#### 4.2.3. S-N Curve

According to the nominal compressive stress of the welded details on both sides of the specimen and the corresponding fatigue life, an S-N curve can be made by taking the inverse slope in the double logarithmic coordinates equal to *m* = 5 according to the British standard BS7608: 2014 + A1: 2015 [27,28], or it can be m = 3 when the load cycles are *N* < 10^7^ according to the American standard API RP 2A [29], taking m = 5 as an example, as shown in Figure 13. For the cracked welded detail, a nominal stress of 54.6 MPa corresponds to a fatigue life of 300,000 cycles, and, for the welded detail on the opposite side of the cracked welded detail, a nominal stress amplitude of 15.2 MPa corresponds to an infinite fatigue life of more than 2 million cycles. Through the comparison between 15.2 MPa and the maximum nominal stress range bridge, it is clear that the welded detail of the external inclined strut joints under the fatigue vehicle load is safe.

## 5. Parametric Analysis and Discussion

### 5.1. Establishment of Refined Submodels

According to the measured results of the strain at the welded detail, the stress state at the welded details of the external inclined strut welded joint was complex, so refined submodels for the bridge’s segment were established, as shown in Figure 14. The most refined submodel was for a region of about 60 mm × 50 mm near the end of the gusset plate where the stress is highly concentrated. The global mesh sizes of the submodels’ levels 1, 2, and 3 were 50 mm × 50 mm, 8 mm × 8 mm × 8 mm, and 2 mm × 2mm × 2 mm, respectively. Moreover, the mesh of the welded detail was further refined, and C3D20R solid elements were used, which are 20-node quadratic solid elements with eight points of numerical integration. The model and submodels included a total of 412,070 nodes and 351,952 elements. All the elements are reduced integration. The static, general step procedure was applied, and the nonlinear effects of geometry were not considered during preprocessing. The calculation showed a good convergence. In this model, the symmetry of the composite box girder section was used, and, therefore, the symmetry boundary was applied. The load was also fatigue load model Ⅲ, and the constraints between the different levels of the submodels were the Tie type.

### 5.2. Calculation Results

As shown in Figure 15, the Mises stress (in MPa) distribution of the model near the welding hole shows an obvious “butterfly”-shaped stress distribution.

Figure 16 shows the calculation results of the maximum principal stress for the welded details of the refined model with a welding hole, the red area is the tensile stress concentration area, and the blue area is the compressive stress concentration area. The results show that there is significant stress concentration around the hole, with compressive stress concentration on both sides of the hole and tensile stress concentration at the bottom and weld ends. The tensile stress at the end of the weld is the largest, which should be related to the stress concentration caused by the obvious geometric mutation.

In order to analyze the intensity of the stress concentration of the joint welding details, the Mises stresses along two paths were extracted. One is the circumferential path passing through the ends of the welding holes, and the other is the longitudinal path passing through the weld toes and the edges of the welding holes, as shown in Figure 17. Figure 18 shows the stress distribution curves along the two paths. It is clear that the welding hole produces a high principal tensile stress under the compressive loading with a large eccentricity. The degree of stress concentration is evaluated by the stress concentration factor *K*, which is defined by Equation (2). The concentration factor calculated by the data of the circumferential path is 5.23.
(2)$$K=\frac{\sigma_{\max}}{\sigma_{\mathrm{ave}}}$$

### 5.3. Parametric Analysis

There is a high probability of fatigue cracking which may cause fatigue damage to the external inclined strut. Therefore, it was necessary to optimize the design parameters of the external inclined strut welded details. Here, the flange thickness (20 mm, 24 mm, and 30 mm), tube thickness (280 mm, 320 mm, 360 mm, and 400 mm), and diameter of the welding hole (30 mm, 50 mm, and 70 mm) were selected for a parametric analysis.

Figure 19 shows the distribution of the principal stresses with the circumferential path in each group of finite element models. The results indicate that the stress near the welding hole changes sharply, while the stress of the rest of the region changes relatively gently. As shown in Figure 19a, with the increase in the tube thickness, the principal stresses (the absolute value of the principal stress, similarly hereinafter) in the weld toe and tube body increase, and the stresses at the hole edges also increase. This is because the principal stresses in the weld toe and strut body are related to both the strut stiffness and nominal stress. The increase in tube thickness leads to an increase in the section modulus and stiffness of the tube, which creates more internal compressive forces in the external inclined strut joint and less internal force in the other parts of the composite box girder. On the contrary, the increase in the thickness of the tube section reduces the nominal stress if the internal forces are the same. In addition, Figure 19a indicates that the effect of the increase in the internal forces caused by the change in section modulus and stiffness is more significant, so the principal stresses increase with the increase in tube thickness. As shown in Figure 19b, when the flange thickness increases, the principal stresses in the weld toe and tube body decrease. The peak value of the principal stresses also decreases, which indicates that the tube stiffness increased by the increase in the flange thickness is not obvious; therefore, the increase in the internal compressive force is not obvious, while the effect of the nominal stress decrease is more obvious. As shown in Figure 19c, with the increase in the diameter of the welding hole, the principal stresses in the weld toe and tube body have no obvious change, while the principal stresses at the edge of the welding hole decrease. This indicates that the effect of the local structural changes on the overall structure is small, and the larger diameter of the welding hole makes the stress concentration lower, so the peak stress at the edge of the hole reduces.

The stress concentration factors *K* of each group of parametric models are shown in Table 2. It can be seen that tube thickness and flange thickness have less influence on the stress concentration factor, while the stress concentration factor increases sharply with the decrease in the diameter of the welding hole. Therefore, it is necessary to adopt a reasonable diameter for the welding hole to control the effect of stress concentration.

## 6. Conclusions

In this study, the fatigue performance of an external inclined strut welded joint was studied. One full-scale fatigue test was conducted. Furthermore, the parametric analysis of the external inclined strut was carried out. The following conclusions can be drawn:

(1) The most unfavorable welded detail of the inclined strut of the wide-flanged composite box girder bridge was the inner flange upper joint of the external inclined strut in the middle span, and the maximum nominal stress range of the welded detail was 6.70 MPa. Meanwhile, the three-dimensional refined finite element model showed that the stress concentration around the welding hole of the external inclined strut welded joint was obvious. Although the overall strut was under compressive loading, the effect of stress concentration of the welding hole made the local principal tensile stress as high as 30.4 MPa, so there was a risk of fatigue damage.

(2) In the full-scale fatigue test of the external inclined strut welded joint, fatigue cracks appeared on the side of the eccentricity and did not appear on the other side. The fatigue life of the most unfavorable welded fatigue detail at the nominal stress range of 54.6 MPa was about 300,000 cycles, and the fatigue life at the nominal stress range of 15.2 MPa was over 2 million cycles. Both the finite element calculation and the full-scale fatigue test showed that the fatigue strength of the external inclined strut welded joint of the bridge was safe.

(3) The results of a parametric analysis showed that the tube thickness and flange thickness had an effect on both the internal force and nominal stress which contributed to the principal stress of the external inclined strut. With the increase in tube thickness, the principal stress increased, but, with the increase in tube thickness, the principal stress decreased. In addition, with the increase in the diameter of the welding hole, the stress concentration factor reduced. In conclusion, the increase in tube thickness and the diameter of the welding hole is beneficial to the fatigue life of the external inclined strut welded joint.

## Figures and Tables

**Figure 1 materials-16-03637-f001:**
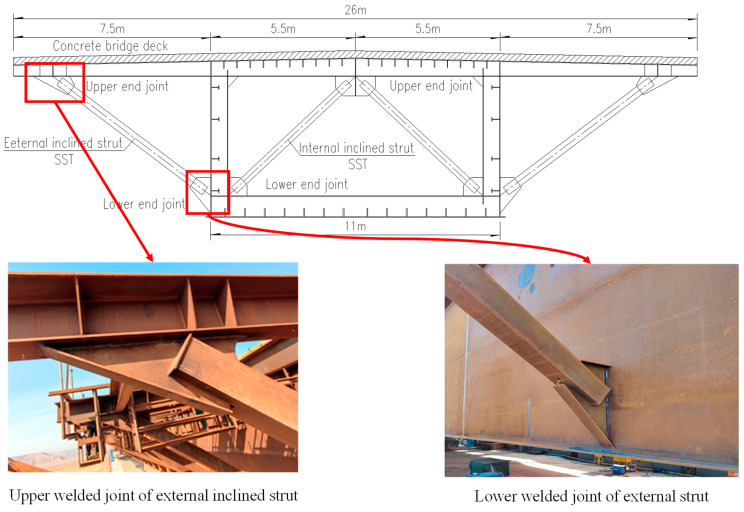
Design of upper and lower welded joints of external inclined strut.

**Figure 2 materials-16-03637-f002:**
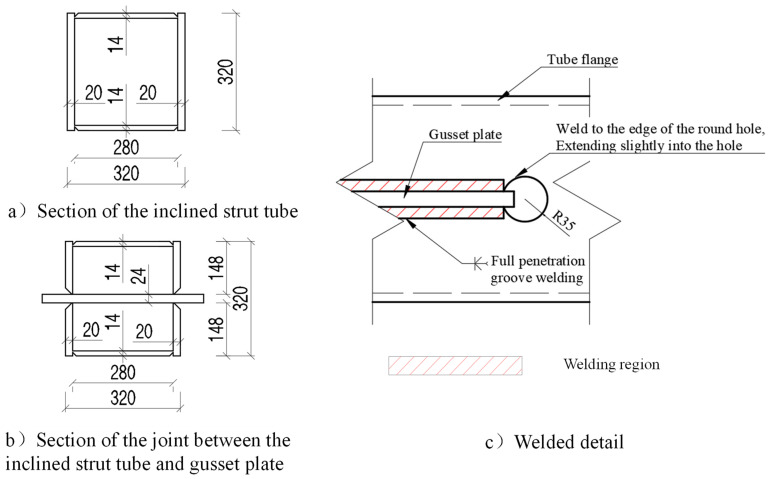
Cross-sectional drawing and welded detail design of external inclined strut.

**Figure 3 materials-16-03637-f003:**
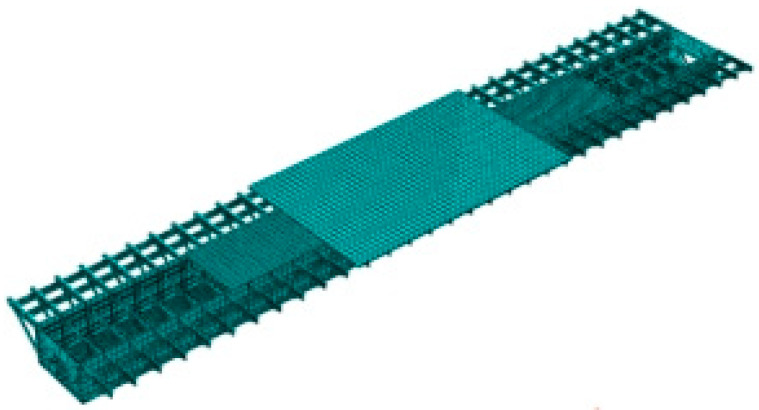
Finite element model of composite box girder.

**Figure 4 materials-16-03637-f004:**
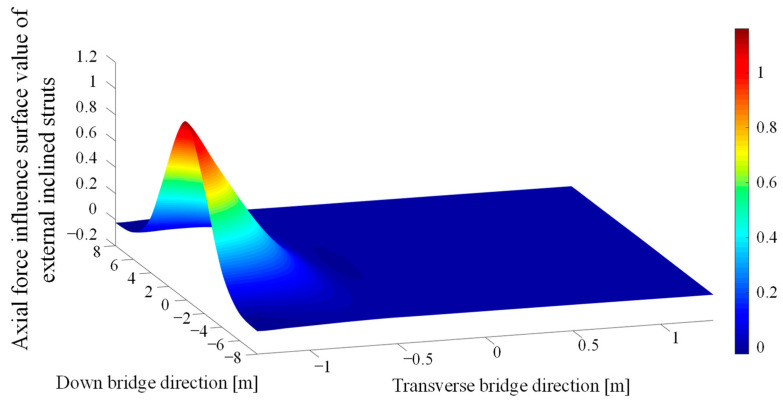
Axial force influence surface at 1/4 span of external inclined strut.

**Figure 5 materials-16-03637-f005:**
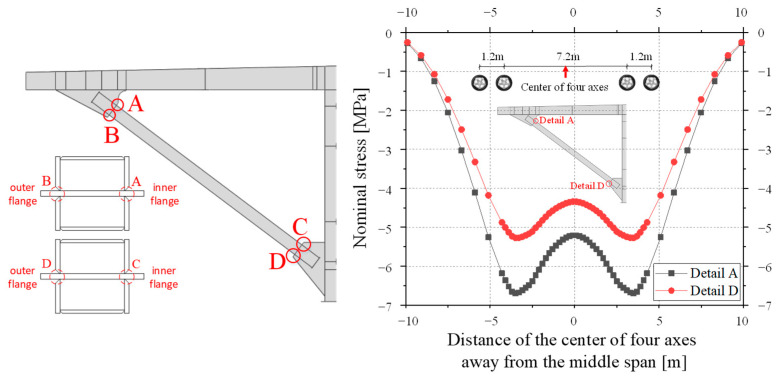
Nominal stress history of fatigue-sensitive details A and D.

**Figure 6 materials-16-03637-f006:**
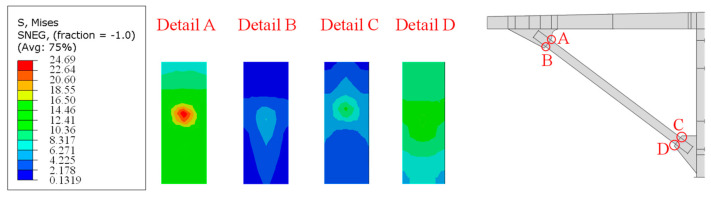
Mises stress distribution of the inside and outside of the external inclined strut from the finite element simulations.

**Figure 7 materials-16-03637-f007:**
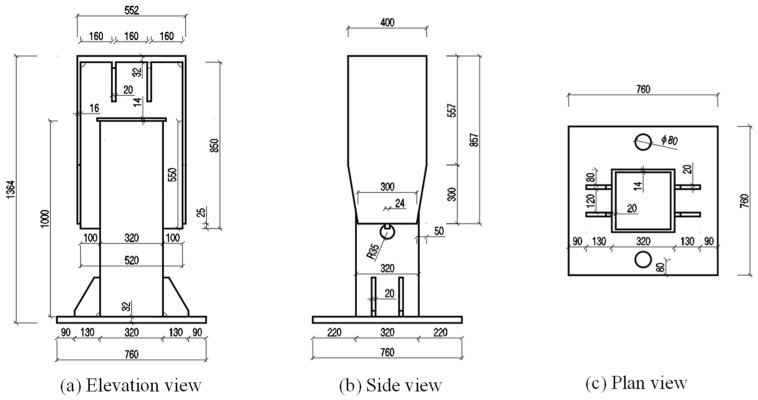
Three views of design of specimen S1.

**Figure 8 materials-16-03637-f008:**
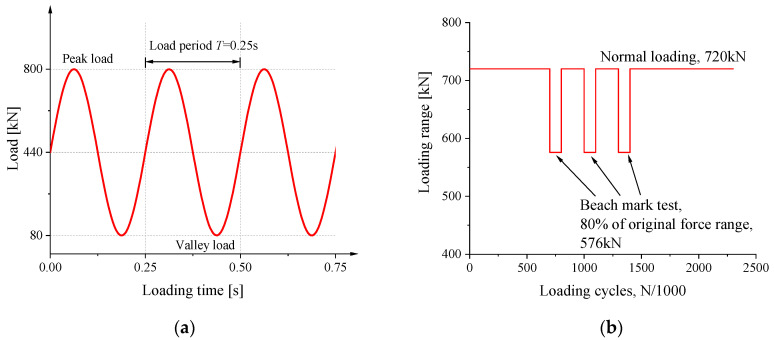
The loading diagram during (**a**) a loading period; (**b**) the whole test.

**Figure 10 materials-16-03637-f010:**
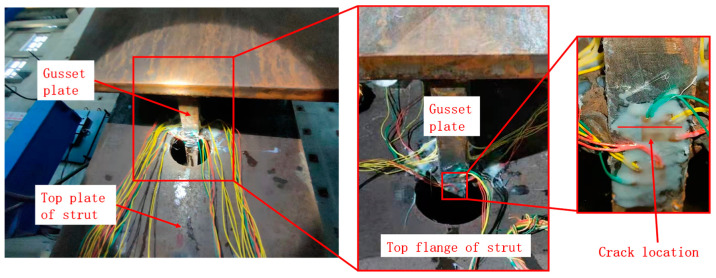
The location of crack.

**Figure 11 materials-16-03637-f011:**
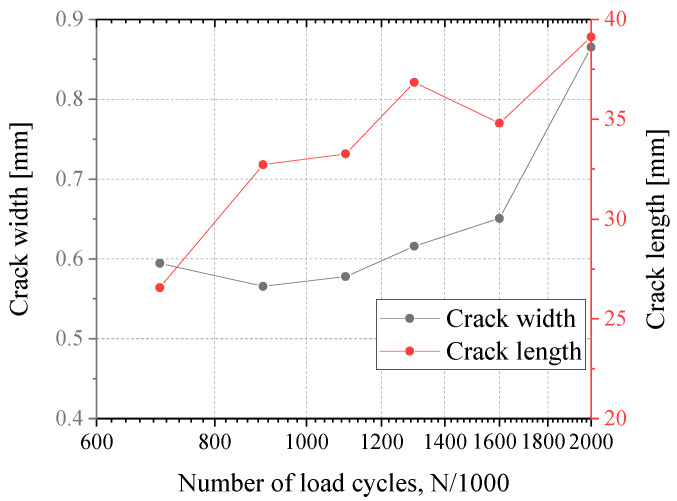
Crack length and width propagation.

**Figure 12 materials-16-03637-f012:**
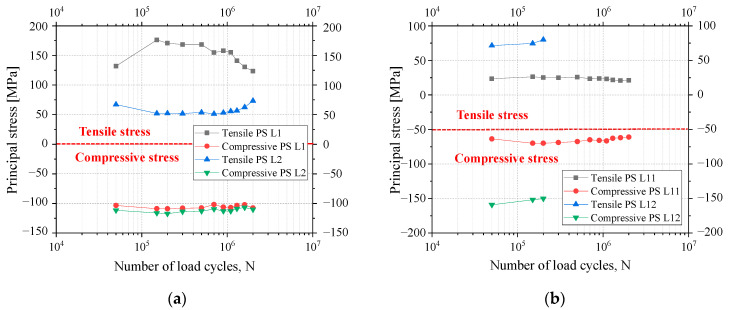
Stresses of the strain gauges near the welding holes on the eccentricity side: (**a**) Principal stresses of strain gauges L1 and L2; (**b**) Principal stresses of strain gauges L11 and L12.

**Figure 13 materials-16-03637-f013:**
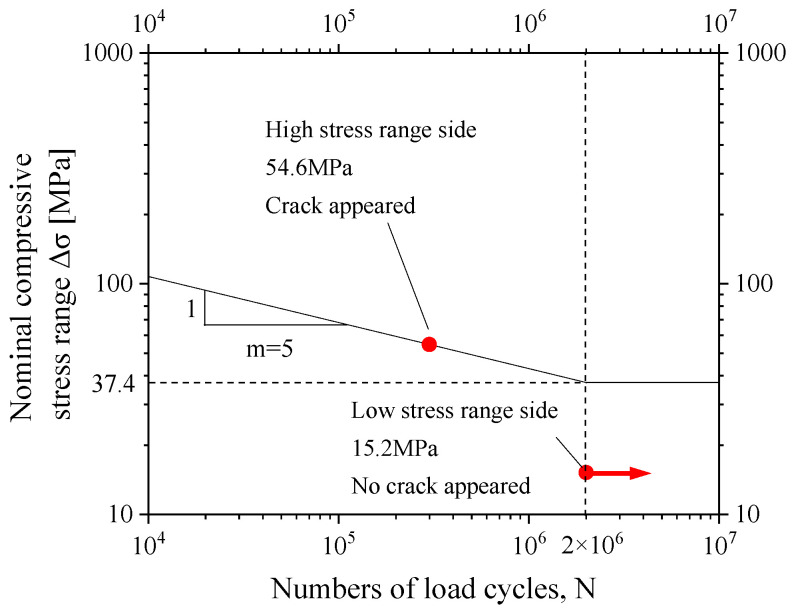
S-N curve based on test results.

**Figure 14 materials-16-03637-f014:**
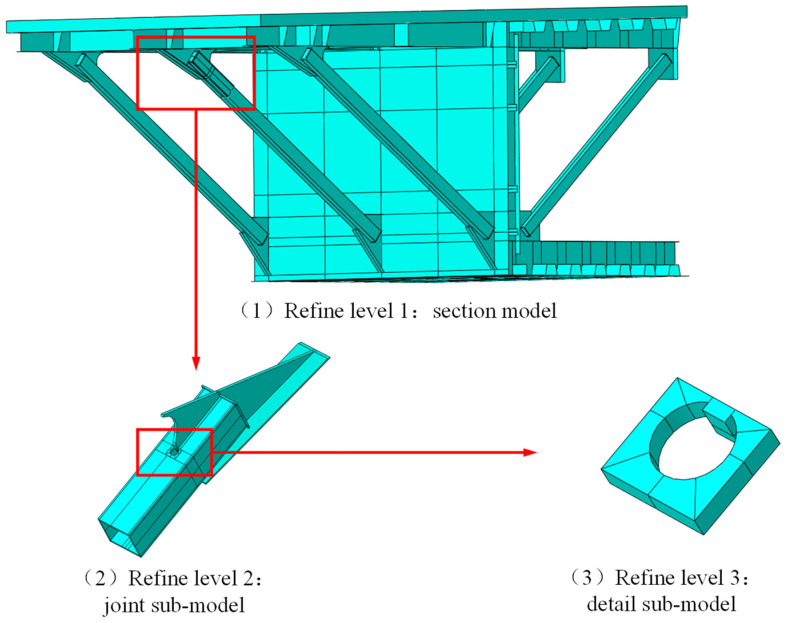
Hierarchically refined submodels.

**Figure 15 materials-16-03637-f015:**
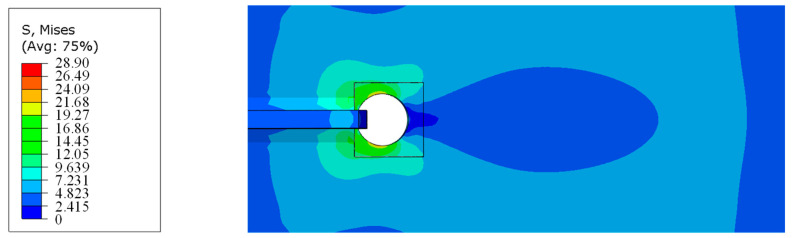
Mises stress distribution of the welding hole.

**Figure 16 materials-16-03637-f016:**
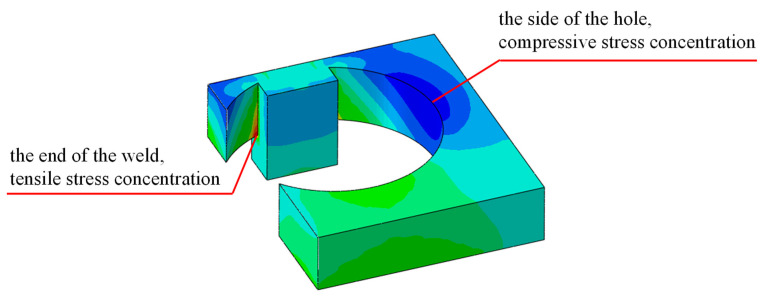
The principal stress distribution around the welding hole.

**Figure 17 materials-16-03637-f017:**
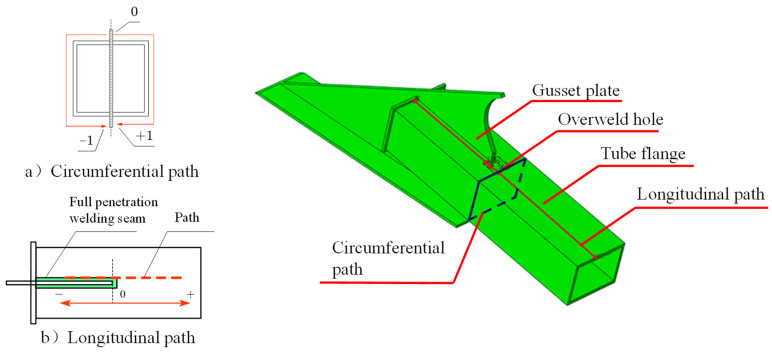
The extraction paths on the external inclined strut welded joint.

**Figure 18 materials-16-03637-f018:**
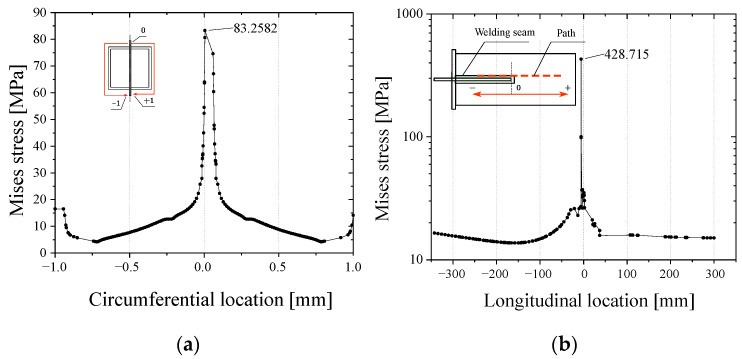
Stress distribution along the: (**a**) circumferential path; (**b**) Longitudinal path.

**Figure 19 materials-16-03637-f019:**
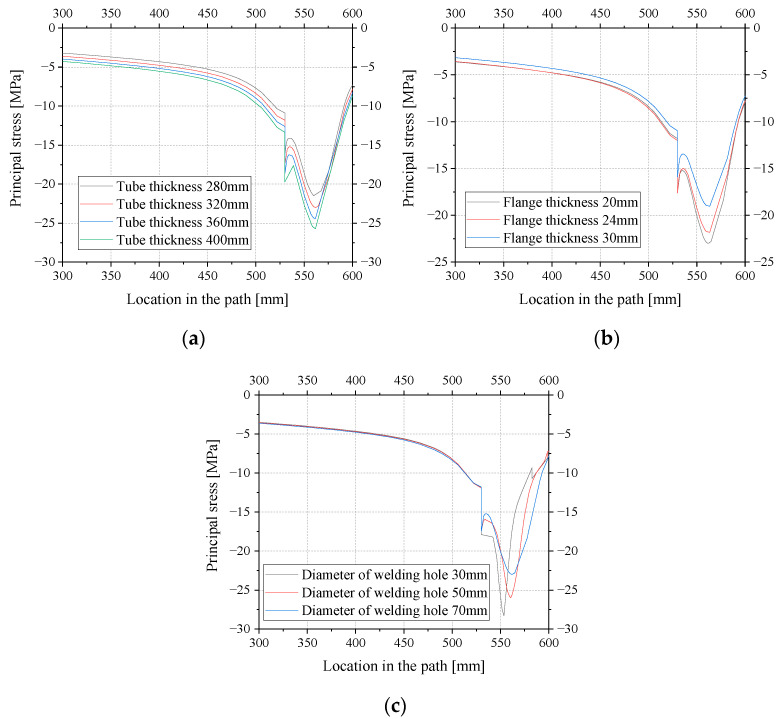
Variation of principal stresses along the load path with parameters: (**a**) tube thickness; (**b**) plate thickness; (**c**) diameter of welding hole.

**Table 1 materials-16-03637-t001:** Material properties of the steel used in the bridge.

Elastic Modulus	Poisson’s Ratio	Yield Strength (20 mm)	Ultimate Strength (20 mm)	Yield Strength (16 mm)	Ultimate Strength (16 mm)
210 GPa	0.3	456 MPa	546 MPa	398	513

**Table 2 materials-16-03637-t002:** Stress concentration factors under different variable parameters.

Tube Thickness (mm)	Stress Concentration Factor *K*	Flange Thickness (mm)	Stress Concentration Factor *K*	Diameter of Welding Hole (mm)	Stress Concentration Factor *K*
280	5.47	20	4.16	30	11.8
320	4.09	24	4.03	50	4.70
360	4.20	30	4.43	70	4.16
400	4.28				

## Data Availability

The data presented in this study are available on request from the corresponding author.

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
