# Peer review of "Full-Scale Fatigue Test and Finite Element Analysis on External Inclined Strut Welded Joints of a Wide-Flanged Composite Box Girder Bridge"

_materials, 2023, doi:10.3390/ma16103637_

Round 1

Reviewer 1 Report

 The paper conducted some experimental and several numerical tests to investigate the stress and fatigue performance of the external inclined strut welded joint. The paper is good. The revision:

a) It should be more disused that how tube thickness and flange thickness have effect on both internal force and nominal stress. Also, the effect of which one is more significant.

b) API RP2A code have several valuable comments for the S-N curve and predicting the fatigue life of welded joints. Add this code.

c) Bellow paper investigated the stress and fatigue life of welded joints. They should be discussed and cited.

Stress concentration factors in tubular T/Y-joints strengthened with FRP subjected to compressive load in offshore structures. International Journal of Fatigue140, p.105719.

Stress concentration factors in tubular T-joints reinforced with external ring under in-plane bending moment. Ocean Engineering266, p.112551.

d) Please add mesh size.

e) We could not see the comparison between the exp. and numerical results. Please clarify this issue.

Reviewer 2 Report

The paper entitled "Full-scale fatigue test and finite element analysis of external inclined strut welded joints of wide composite box girder bridge" bears a good idea and can be accepted for publication after responding to the following comments.

 Comments:

Abstract

1.     The abstract section should be one paragraph up to 250 words, summarizing the main aspects of the entire paper in a specified sequence that includes: 1) the research problem; 2) the general purpose of the research; 3) the methodology and/or data used; 4) The most important findings or results of the research. Hence, please rebuild the summary accordingly.

1. The main objective of the research should be shown at the end of the introduction. 2. What has been done in previous research in this field and what distinguishes this research from it should be mentioned. 3. Discussion of the results requires a more in-depth scientific interpretation and the support of relevant references.

Introduction

1-     It is better to limit the discussion to this point at the beginning of the introduction only and then talk about the methods used in this study and the previous studies related to it, in order to reach the strengths and weaknesses of each analysis method in order to finally clarify the importance of what the authors will do in the current study.

2-     It is not preferable to add references in a combined form. It is preferable to mention the achievements of each researcher separately.

Materials and Experimental Program

1-    All standard specifications that were used must be added.

The paper entitled "Full-scale fatigue test and finite element analysis of external inclined strut welded joints of wide composite box girder bridge" bears a good idea and can be accepted for publication after responding to the following comments.

 Comments:

Abstract

1.     The abstract section should be one paragraph up to 250 words, summarizing the main aspects of the entire paper in a specified sequence that includes: 1) the research problem; 2) the general purpose of the research; 3) the methodology and/or data used; 4) The most important findings or results of the research. Hence, please rebuild the summary accordingly.

Introduction

1-     It is better to limit the discussion to this point at the beginning of the introduction only and then talk about the methods used in this study and the previous studies related to it, in order to reach the strengths and weaknesses of each analysis method in order to finally clarify the importance of what the authors will do in the current study.

2-     It is not preferable to add references in a combined form. It is preferable to mention the achievements of each researcher separately.

Materials and Experimental Program

1-    All standard specifications that were used must be added.

Reviewer 3 Report

The paper must necessarily include nomenclature - a complete list of abbreviations, symbols, markings, etc. - please complete the manuscript.

Introduction is correct - very good literature review - however, please check item citations - spaces are missing in several places before the cited reference - eg lines 30, 34, 40 etc. Please check the manuscript for this.

Figure 1 needs to be enlarged - especially the technical one. The figure must be legible.

Figure 2 should be corrected in accordance with the applicable rules of technical drawing. Please change it - it needs to be more readable. We use thick lines and thin lines - the authors need to change that.

Please add to the thesis a table with information on any mechanical constants, materials from which the bridge was built (Young's modulus, Poisson's ratio, yield strength, tensile strength, etc.). This must necessarily be stated.

Please expand the part of the paper devoted to FEM modeling. Yes, the authors give the sizes of the elements - but these are single sizes, the shell has two dimensions, and 3D elements have three dimensions. More details please. How many points of numerical integration were there in the finite element, what was the type of interpolation in the finite element - were they linear or non-linear elements? How many finite elements were in the model and how many nodes? Please show close-ups of the FEM model in critical places, indicate the boundary conditions, fastenings, load. What is the load on the structure and where is it applied? This must be clearly defined and shown. What is the load pattern? Please complete it. Please provide more details about pre-processing in the calculations. What type of solver did the authors use? Did the authors use the existing axes of symmetry in modelling? How was the convergence of the FEM model assessed? What can the authors say about it? Please complete the paper.

In the figures presenting the results, there is no space before the units given in round brackets - I would change them to square brackets.

Figure 4 should be colored. Please correct.Figure 4 should be colored. Please correct.

In what units are the stresses shown in Figure 6? What do the authors understand by the term "Mises stress"? The authors should clearly indicate that these are effective stresses, calculated according to a specific formula - there is no formula in the manuscript. There is no unit of these stresses. The distributions shown, marked as "Detail A - D" do not contribute anything to the paper, it must be shown for the entire node - single projections of stress isolines do not say anything. It is not known what material these elements are made of - this must be stated in order to be aware of the level of stresses related to the yield point, if we consider effective stresses - the authors do not write anything about the yield point - so the reader cannot refer to the strength of the structure here.

Figure 7 should be corrected - this is supposed to be a clear engineering drawing. We use thick lines and thin lines - the authors need to change that.

Please add to the paper a figure presenting the load diagram in laboratory tests as a function of time.

Please write how many tests the authors conducted in the laboratory? One - on the basis of one test, no conclusions can be drawn. Fatigue tests are subject to a significant scatter - the number of tests must be appropriate so that the results are not accidental.

Please check the figures and in the axis descriptions add spaces before the units given in parentheses.

How did the authors measure the "Crack length and width propagation" shown in Figure 10? Please add details.

Notes on FEM modeling given above should also be referred to section 5.1 in the manuscript. Please correct this section according to the comments above.

The manuscript has potential, but needs major improvements. I recommend a major revision.

Minor editing of English language required.

Round 2

Reviewer 1 Report

ok

Reviewer 3 Report

The authors, with their comments and responses to my comments, convinced me of the importance of this manuscript.

The reviewed article is really very good. The contribution of the authors to its creation should be appreciated.

The authors took into account all my corrections. I accept their responses to my comments.

I recommend the manuscript for publication.

Minor editing of English language required.